# The World Association against Infection in Orthopaedics and Trauma (WAIOT) procedures for Microbiological Sampling and Processing for Periprosthetic Joint Infections (PJIs) and other Implant-Related Infections [note 1]

**DOI:** 10.3390/jcm8070933

**Published:** 2019-06-28

**Authors:** Lorenzo Drago, Pierangelo Clerici, Ilaria Morelli, Johari Ashok, Thami Benzakour, Svetlana Bozhkova, Chingiz Alizadeh, Hernán del Sel, Hemant K Sharma, Trisha Peel, Roberto Mattina, Carlo Luca Romanò

**Affiliations:** 1Clinical Microbiology, Department of Biomedical Sciences for Health, University of Milan, 20100 Milano, Italy; 2Laboratory of Clinical Microbiology, AO Legnano Hospital, AMCLI, 20025 Milano, Italy; 3Residency Program in Orthopaedics and Traumatology, University of Milan, 20100 Milano, Italy; 4Department of Paediatric Orthopaedics and Spine Surgery, Children’s Orthopedic Centre, Mumbai 230532, India; 5Zerktouni Orthopaedic Clinic, Casablanca 20000, Morocco; 6Department of Prevention and Treatment of Wound Infection, R.R. Vreden Russian Research Institute of Traumatology and Orthopaedics, 33701 S. Petersburg, Russia; 7Traumatology & Orthopedics Department, Baku Health Clinic, 1005 Baku, Azerbaigian; 8Department of Orthopaedics, British Hospital Buenos Aires, B1675 Buenos Aires, Argentina; 9Hull University Teaching Hospitals, Anlaby Road, Hull HU3 2JZ, UK; 10Department of Infectious Diseases, Monash University, Melbourne VIC 3004, Australia; 11Department of Odontoiatric and Surgical Sciences, University of Milan, 20100 Milano, Italy; 12Studio Medico Cecca-Romanò, corso Venezia, 2, 20121 Milano, Italy; 13Romano Institute, Rruga Ibrahim Rugova, Tirane 1001, Albania

**Keywords:** microbiology PJI, infection, PJI, peri-prosthetic joint, implant-related, biofilm, diagnosis, definition, pathogen, WAIOT

## Abstract

While implant-related infections continue to play a relevant role in failure of implantable biomaterials in orthopaedic and trauma there is a lack of standardised microbiological procedures to identify the pathogen(s). The microbiological diagnosis of implant-related infections is challenging due to the following factors: the presence of bacterial biofilm(s), often associated with slow-growing microorganisms, low bacterial loads, previous antibiotic treatments and, possible intra-operative contamination. Therefore, diagnosis requires a specific set of procedures. Based on the Guidelines of the Italian Association of the Clinical Microbiologists (AMCLI), the World Association against Infection in Orthopaedics and Trauma has drafted the present document. This document includes guidance on the basic principles for sampling and processing for implant-related infections based on the most relevant literature. These procedures outline the main microbiological approaches, including sampling and processing methodologies for diagnostic assessment and confirmation of implant-related infections. Biofilm dislodgement techniques, incubation time and the role of molecular approaches are addressed in specific sections. The aim of this paper is to ensure a standardised approach to the main microbiological methods for implant-related infections, as well as to promote multidisciplinary collaboration between clinicians and microbiologists.

## 1. Introduction

Periprosthetic joint infection (PJI) remains a serious complication in orthopaedic surgery. The infection rate after primary knee or hip replacement is estimated to range between 0.3 and 1.9% [1], and may exceed 10% in revision surgery or in patients with specific risk factors [2]. Similarly, the incidence of infection after internal osteosynthesis (intramedullary nails, plates, screws, etc.) ranges from 1 to 2% for closed fractures to over 30% for the open fractures. 

Pathogen identification through microbiological analysis of samples is pivotal to confirm the diagnosis of PJI. Microbiological findings are a common criteria incorporated into definitions of PJI and diagnostic algorithms, including the definition recently released by the World Association against Infection in Orthopaedics and Trauma (WAIOT) (Table 1 and Table 2) [3,4,5]. 

The microbiological diagnosis plays a central role in infection confirmation, and also is critical for determining antimicrobial susceptibility of the pathogen(s) to guide the antimicrobial treatment.

Microbiological diagnosis of implant-related infection has unique aspects that may make it extremely challenging. Firstly, the presence of bacterial biofilms in virtually all implant-related infections, and especially in chronic infections, coupled with the ability of the microorganisms to persist in a slow-growing or even intra-cellular state, markedly reduce the sensitivity of traditional microbiological culture techniques for pathogen(s) detection and identification. The presence of biofilms plays a role in “acute” and “chronic” clinical presentations. Biofilm formation is known to occur in few hours after bacterial adhesion on a surface. The clinical features of acute, subacute and chronic infections are related to the host’s response and to the interaction between the bacteria and the host. The age and maturity of the biofilm is an important factor influencing diagnostic yields: in established, “chronic” infections, lower sensitivity of diagnostic tests is noted. This may relate to low bacterial loads, slow-growing states and prior antibiotic exposure [6,7]. Added to these challenges, microbiological sampling undertaken during complex and prolonged surgical procedures, may lead to bacterial contamination of the specimens and false positive results.

Despite of the key role of microbiology culture techniques to diagnose these complex implant-related infections in orthopaedics and trauma, there is a universal lack of standardized and shared procedures for microbiological sampling and processing. According to a recent survey conducted in twenty leading orthopaedic centres, a remarkably heterogeneous spectrum of microbiological procedures was reported across the world [7]. This impacts on the overall diagnostic accuracy and reproducibility of the results and has prompted the World Association against Infection in Orthopaedics and Trauma to draft the present document, based on the Guidelines of the Italian Association of the Clinical Microbiologists (AMCLI) released in 2013 and revised in 2017, on “Microbiological Sampling and Processing of Implant-Related infections” [8].

## 2. Microbiological Features and Actual Issues 

Although still debated, PJIs can be diagnosed when at least one of the following criteria is present: two positive cultures with microorganisms having identical phenotype, or the presence of a sinus tract in communication with the prosthesis [3,4,9,10,11,12]. Alternatively, at least three of the following five minor criteria can be satisfied to establish the diagnosis of PJI: high ESR and CRP [13,14,15]; elevated leukocytes count in joint fluid [9,16,17]; “++” positive test of leukocyte esterase on a test strip performed on joint fluid [18,19]; elevated percentage of neutrophils in joint fluid [9,16,17]; acute inflammation of periprosthetic tissues at histological examination [20] and/or; a positive culture in a single sample. Despite these criteria, there is a lack of an overall agreement between the different microbiological laboratories and countries on optimal diagnostic approaches, reflecting limited collaboration between Microbiologists and Orthopaedic surgeons.

The site of intraoperative periprosthetic tissue sampling influences the yield of microbiological culture for periprosthetic joint infections. Bacteria can infect different sites of the prosthetic components as well as in the tissues. The rate of positivity of samples differs when specimens are obtained from tissues or fluids in contact with prosthetic material compared with cancellous bone [21]. However, the rate of positivity is also dependent on the modality of collection, transportation and processing of samples, as well as on the type of microorganism and the host’s response. The presence of low-grade pathogens, such as coagulase-negative staphylococci, *Enterococcus* spp, *Corynebacterium* spp, *Cutibacterium acnes*, can further impact the diagnostic yield when compared with highly pathogenic microorganisms, such as *Staphylococcus aureus*. In the setting of a mature and established biofilm, many common microbiological culture techniques may be falsely negative, further compounding laboratory diagnostic challenges. 

Staphylococci are the most frequently isolated microorganisms in both early and late infections [22]. Coagulase negative staphylococci, such as *Staphyloccocus epidermidis*, *Staphyloccocus lugdunensis*, *Staphyloccocus capitis*, *Staphyloccocus hominis*, *Staphyloccocus caprae* are the most common organisms isolated in PJIs (30–43%), followed by *S. aureus* (12–23%) [23,24,25,26]. Streptococci, enterococci and diphtheroids are isolated in about 10% of cases, and gram-negative bacteria (Enterobacteriaceae and Pseudomonas) in about 8% [25]. Among the anaerobic bacteria, *Cutibacterium* (former *Propionibacterium*) *acnes* is the most frequent, while 10–12% of infections are caused by more than one microorganism [26]. In the haematogenous infections, *S. aureus* and coagulase-negatives are the predominant microorganisms, followed by *Streptococcus* spp and Gram-negative bacteria [27]. 

The infections related to the osteosynthesis devices are mainly caused by *S. aureus*, but are often polymicrobial [28]. 

Other microorganisms may be also involved after a direct inoculation or haematogenous spreading, and the aetiology includes mycobacteria and fungi [29]. 

Often the diagnosis of infection is complicated by the failure of traditional microbiological culture approaches to isolate the causative pathogen. This is due to the fact that microorganisms are organized and embedded within complex structures, known as biofilms, which typically occur on the surface of prosthetic material. These biofilm associated microorganisms are difficult to diagnose and treat [30]. The adhesion of microorganisms to prosthetic surfaces reduces their detection. An additional issue is the presence of “small colonies variants”, a distinct bacterial phenotype that may occur in the microbial sub-populations within the biofilm. The slow growth of this microbial variant may also increase the time of detection and impacts clinical management outcomes. 

Currently methods to evaluate the in vitro activity of antimicrobials against biofilm-associated microorganisms are not routinely available in clinical laboratories [29].

## 3. Methodology

These procedures have been developed through systematic review of recent publications. An electronic search of all available literature included the following search term “Microbiology procedures for the diagnosis of Prosthetic Joint infections”. 

The search included digital sources on PubMed (http://www.ncbi.nlm.nih.gov/pubmed/), ScienceDirect (www.sciencedirect.com), Institute of Scientific Information (ISI) Web of Knowledge (http://www.isiwebofknowledge.com) and Google Scholar (http://scholar.google.com). This review also incorporates manual review of included publications on the website of the Italian Scientific Society of Clinical Microbiology (AMCLI). Only peer-reviewed and highly impactful Journals were considered. No typical microbiological methods or tests were excluded in this analysis. 

The following sections summarize and critique microbiological issues along the diagnostic pathway, from samples collection to culture processes and interpretation in the microbiology laboratory. 

### 3.1. Microbiological Sampling 

The culture and isolation of the microorganism is a major criterion for the diagnosis of prosthetic joint infections. Samples suitable for microbiology testing are: periprosthetic tissues, joint fluid and/or, prosthetic components removed during the revision procedure. 

During the pre-operative phase, culture of joint fluid can aid diagnosis and is recommended [31]. Joint fluid collection should be performed by arthrocentesis and the microbiological culture of joint fluid must include approaches to detect aerobic and anaerobic microorganisms (see below). 

Blood cultures for aerobic and anaerobic microorganisms also may be performed, particularly in patients with fever and/or acute onset of symptoms, or a concomitant infection that makes the presence of bacteraemia highly probable, such as infective endocarditis [4,32]. 

Percutaneous biopsy of peri-implant tissues, preferably supported by ultrasound guidance or other imaging systems, such as fluoroscopy, may be suitable for microbiological aerobic and anaerobic cultures. In case of prosthetic loosening, the biopsy may be taken from the cement-bone interface or around the prosthetic components. Collection of an additional periprosthetic tissue sample also may be useful for histological examination [29]. In the intra-operative phase, microbiological investigations should include cultures of joint fluid or of the joint capsule. Aerobic and anaerobic cultures must be performed on periprosthetic tissue and joint fluid specimens. A minimum of four periprosthetic tissue samples should be collected. Alternatively three periprosthetic tissue specimens maybe adequate if the homogenate cultures are furtherly inoculated in blood culture bottles [33]. Retrieved prosthetic components and/or osteosynthesis devices can be processed using specific methods, such as sonication, to dislodge bacteria from the biofilms and the sonicate fluid should be cultured aerobically and anaerobically. Samples of external fixators or any purulent material collected around the fixator pin by a sterile syringe may be also collected. However, sampling of external fixator pin site in clinical setting has limited value, due to the risk of contamination of the specimen with skin flora. Therefore this practice should be discouraged [11].

Guidance for intra-operative specimen processing, from the sample collection until the microbiological report, is described below.

#### 3.1.1. Sample Collection

The collection of synovial fluid is performed by percutaneous joint aspiration using aseptic technique, and may be performed with ultrasound guidance, particularly for aspiration of the hip joint. The fluids can be inoculated onto appropriate culture media or blood culture bottles (≥1ml per bottle) for aerobic and anaerobic microorganisms [21,34,35]. The remaining aspirate should also be transferred into sterile vials for subsequent cultures and in ethylenediaminetetraacetic acid (EDTA) tubes for total leukocyte and polymorphonuclear neutrophil count. In osteosynthesis device infections syringe aspiration of fluid collections in the area surrounding the implant may be helpful.

Intraoperatively, the collection of multiple biopsies is essential to increase the sensitivity of culture methods and to distinguish contaminating microorganisms from pathogens. Each biopsy should be collected using separate, sterile instruments. Periprosthetic fields should be explored and sampled, with careful attention to avoid contact with the skin and other external materials of operative field [3,21]. Biopsy should then be inserted directly into sterile and hermetically sealed tubes or containers to prevent any risk of contamination. Three to six biopsies should be collected [4]. In case of prosthetic infections, the sampling should preferentially include biopsies from the implant-bone interface or from the joint capsule or any areas with macroscopic signs of inflammation [21]. In case of osteosynthesis devices infections, the samples should be collected from the peri-implant inflamed area. A volume of approximately 1 cm^3^ for tissue biopsy samples is recommended to aid analysis, including the histopathological examinations [20]. 

All the explanted prosthetic components (femoral stem, acetabular cup, femoral shield, tibial plate, polyethylene insert, humeral stem, glenoid), the periprosthetic cement, or the osteosynthesis devices (screws, plates, intramedullary nails) as well as the non-absorbable sutures or the bone substitutes, are considered suitable samples for microbiological examination. The probability of finding bacteria organized in their biofilms is very high in these samples. Particular attention must be paid during the collection of prosthetic/biomaterial components to avoid contamination. Components should be excluded in the event of direct contact with the patient’s skin during extraction/removal, or if instruments used are suspected to have had direct contact with the skin. Components should be transferred into a non-perforable, sterile, leak-proof container of suitable size, and care should be taken to minimize further manipulation by the operators. The containers must be properly sealed and sent to the laboratory as soon as possible. To increase the specificity of the culture each component should be placed in a separate container [36]. If the components are to undergo sonication for detachment of microorganisms organized in biofilm, it is advisable to cover the at least 90% volume of the removed component with Ringer’s solution or sterile physiological solution, avoiding external microbial contaminations [36,37,38,39]. It is also recommended to send an additional sterile, non-perforable containers with or without Ringer’s solution or sterile physiological solution, to act as a negative control for culture following the same procedures used for explanted prosthetic components [40].

#### 3.1.2. Transportation

Transportation of samples to the laboratory must performed in a timely manner. Tissue samples and prosthetic components should be stored at 4 °C if transport to the laboratory is delayed [29]. Blood cultures bottles inoculated with joint fluid can be stored at room temperature up to 48 h. To aid communication, the following information should be included with the specimens: date and time of collection, name of a contact person collecting the samples, anatomical site and any clinical information (antibiotic therapy, previous infectious diseases), specifying any request for additional microbiological tests (i.e. mycobacteria) [11]. 

#### 3.1.3. Swabs and Drainage fluids

Sinus tract swabs or secretions are not so suitable for cultures because isolated microorganisms often represent the colonizing microbial skin flora. Organisms isolated from sinus tract swabs correlate poorly with the etiologic agent responsible of a deep infection, with the only exception of *S. aureus* [26]. Swabs of peri-prosthetic materials should be avoided as the culture sensitivity is low, compared with tissue samples or prosthetic components [41,42]. The preparation of cultures from drainage or drainage liquids is also not recommended due to the high risk of contamination [11]. Similarly swabs of open wounds are discouraged.

### 3.2. Microbiological Culture Approaches 

#### 3.2.1. Samples Handling and Incubation

The materials used to diagnose prosthetic or osteosynthesis device infections must be handled in a class 2 biological cabinet. To minimize the possibility of contamination the number of times in which the laboratory staff handle the samples or open containers should be limited [29]. The technician must use disposable gloves (and replaced gloves during prolonged work). Specimen inoculation and inspection for microbiological growth on culture media must be carried out under the same conditions [11]. Culture methods include the use of solid agar plates as well as broth media for enrichment [29,34]. Incubation should be a minimum of 5–7 days for aerobic cultures and up to 14 days for anaerobic culture methods. Prolonged incubation is recommended when chronic and delayed infections due to slow-growing microorganisms and anaerobic bacteria, such as *Cutibacterium acnes*, are suspected, for example with prosthetic shoulder joint infections [21,43,44,45]. Several studies in recent years report the advantages of using blood culture bottles for culture of joint fluid, periprosthetic tissue homogenate and sonication fluid from the prosthesis. The inoculation of blood culture bottles allows continuous growth monitoring as well as the possibility of inoculating larger sample volumes than solid media. The presence of antimicrobial removal systems and lytic agents in the blood bottles promotes in addition the release of intracellular microorganisms [21,29,34,46,47,48,49,50,51]. By using blood culture and semiautomatic monitoring systems, a reduced time to culture positivity has been observed, when compared to conventional agar plates and broth methods [46,50,52]. The bottles should be incubated for a minimum of 7 days for the aerobic bottle and 14 days for the anaerobic bottle [46,51,53]. A Spanish group experienced that extending incubation of the samples to 14 days does not add more positive results for sonicated orthopaedic implants compared with a conventional seven-day incubation period [54].

The use of blood cultures bottles for inoculation of joint fluids directly at the patient’s bedside, during pre-operative or operative revision surgery, may also assist with the diagnosis of chronic infections, or in patients exposed to antibiotic treatments prior to sampling [11,29].

In acute infections, a Gram stain of the joint aspirate can be also useful, although a negative result does not exclude the possibility of infection [29]. In general, Gram staining has high specificity, but low sensitivity (Sensitivity 26%, Specificity 97%) [55]. Cultures of tissues can be performed with sterile solution 0.1% (w:v) of dithiothreitol or by homogenizing the biopsies in 3 ml of broth (i.e., Brain Heart Infusion broth) in aseptic environment [56,57].

Literature suggests biopsies should undergo pre-treatment with homogenization. A number of different approaches to homogenization exist [21,33,58,59,60]. However, anecdotal reports have suggested challenges with the homogenization procedure, therefore vortexing may be used instead for specimen preparation.

#### 3.2.2. Sonication or Dithiothreitol (DTT) Biofilm Dislodging Procedures

Sonication improves the sensitivity of culture of prosthetic components or osteosynthesis devices [39,57,61]. The containers with prosthetic components must be handled under a laminar flow cabinet and covered to at least 90% of its volume with Ringer’s solution or sterile physiological solution. The samples are vortexed for 30 s, sonicated at 30-40 KHz 0.22 ± 0.04 W/cm^2^ for 5 min and vortexed again for additional 30 s. These procedures must be performed with careful attention to sterile technique to minimize possible contamination. Other authors include a further step of 5 min of centrifugation to improve the detection and the sensitivity of sonication. Sonication may be also useful in other specific conditions, such as pedicle screw loosening in spinal surgery and infection diagnosis on megaprostheses [62,63]. Furthermore, even if limited evidence exists to date, sonication could be used on cement spacers, when a persistent infection is suspected [64]. 

The use of a solution of dithiothreitol (DTT) may be used as an alternative to sonication [65]. In this approach a sterile solution of 0.1% (w: v) of dithiothreitol (DTT, formula C_4_H_10_O_2_S_2_, molecular weight: 154.2) in phosphate buffer saline (PBS) is added to cover the prosthetic components. The container with prosthetic components and DTT solution is shaken up at about 80 rpm for 15 min. The incubation approaches for aerobes and anaerobes are in keeping with the previously described methods. A recent study found comparable results between DTT and sonication for the detection of PJIs in 232 patients undergoing revision, and both approaches were more sensitive than standard tissue cultures [66].

Each method used by the laboratory should be properly certified to ensure standard approaches to microbiological culture and microorganism identification. 

#### 3.2.3. “Atypical and Rare” Microorganisms

Attention should be paid to the “small colonies variant” and to the microorganisms removed from the biofilm as they may represent aberrant phenotypes which result from inactivation of specific enzymatic processes. However, the repeated subcultures of these “small colony variants” in enriched media often lead to these variants returning to their original phenotypic traits [67,68].

Microorganisms, such as *Abiotrophia defectiva* and *Granulicatella adiacens* (previously classified as a nutritional variant of streptococci) are difficult to detect in culture. Additional cultures for fungi and mycobacteria should be considered in patients with clinical evidences of prosthetic infection in whom standard microbiological cultures are negative. Additional molecular tests may be performed to allow detection and identification of non-culturable organisms (including antimicrobial affected organisms), or serological tests for Brucella and Coxiella [69]. Infections caused by the so-called low-grade pathogen still may be present in patients with no specific symptoms or sub-clinical signs (absence of fever, negativity of inflammation or infection markers, absence of sinus tracts). In these cases, C-Reactive Protein (CRP) and Erythrocyte Sedimentation Rate (ESR) values may not to be markedly increased. In patients with presumed aseptic failure of the prosthesis, dedicated molecular methods have resulted in the detection of a microorganism in 4–13% of cases [70]. These detected pathogens are usually low virulence organisms, and include coagulase-negative staphylococci, cutibacteria, corynebacteria. These infections are typically characterised by the absence of host inflammatory responses and tissue damage which decreases the diagnostic sensitivity of standard clinical evaluations or common laboratory parameters [30]. In these sub-clinical infections, joint fluid cultures may be negative, therefore sample processing assumes particular relevance to augment their detection [30,71,72].

#### 3.2.4. Antimicrobials for Susceptibility Testing

The following antimicrobials may be tested in vitro according to the EUCAST guidelines (www.eucast.org), guided by the type of bacterial identification: rifampicin, levofloxacin, moxifloxacin, ciprofloxacin, daptomycin, teicoplanin, vancomycin, linezolid, minocycline, clindamycin, fusidic acid, cotrimoxazole, amoxicillin/clavulanic acid, piperacillin-tazobactam, meropenem, ertapenem, oxacillin, fosfomycin, 3rd generation cephalosporins (such as ceftriaxone), colistin and, tigecycline. The Minimum Inhibitory Concentrations (MICs) should be reported. 

#### 3.2.5. Molecular Methods

Culture remains the gold standard in the diagnosis of periprosthetic joint infections. Molecular techniques have a role in some specific cases such as negative cultures with a strong suspicion of infection, or in case of fastidious microorganisms [73,74]. Molecular techniques can confirm the presence of microorganisms and provide their identification, but they do not give a full and exhaustive antibiotic-sensitivity profile for all the antimicrobials indicated for PJIs therapy. The sensitivity of the molecular tests is dependent on sample type [73,75,76,77,78]. At present, molecular tests have not been fully incorporated into routine laboratory diagnostic protocols for PJI because of the high costs of these tests and lack of data to support the superiority of polymerase chain reaction (PCR) compared with traditional culture methods [79,80]. Recently studies examining multiplex-PCR or microarrays have reported high specificity (greater than 95%) [81]. However, primers for the common PJI isolates (such as *Cutibacterium acnes* may not be included in commercially available multiplex PCR kits [82]. Metagenomic shotgun sequencing may be another interesting tool for diagnosing prosthetic infections. Ivy MI et al. demonstrated that this approach can detect pathogens involved in PJI when applied to culture-negatives synovial fluids [83]. 

Therefore, ongoing research and development of non-cultural methods (microarray, next generation sequencing) may realize improvements in sensitivity and diagnostic accuracy, as well as shortening the detection time of pathogens, and standardizing and automating analytical procedures. 

### 3.3. Reporting of Microbiological Results 

Reporting an accurate diagnosis of infection in the setting of prostheses or fixation devices requires an integrated approach between laboratory capabilities and the clinical results. This approach could also help to define the potential etiological role of those microorganisms isolated from a single tissue sample or prosthetic component [38]. The growth of a virulent microorganism, such as *S. aureus*, *Pseudomonas aeruginosa*, Enterobacteriaceae, or anaerobic bacteria, in a single tissue biopsy or prosthetic/joint fluid sample may be consistent with infection [4,21]. The growth of microorganisms, such as coagulase-negative staphylococci and *Cutibacterium acnes*, from a single sample may not be indicative of infection and may represent contamination of the specimen. The specific role of the low-grade microorganisms should be evaluated by the clinician on the basis of clinical and investigation findings [4]. These microorganisms should be reported as “Possible contaminating microorganism: assess its clinical significance”. Microbiological reports should be performed with timely communication of clinically significant positive cultures, and should contain the microorganism/s identification and the relevant antibiotic susceptibility tests [29]. It may be of assistance to the clinicians if the microbiology laboratory provides a preliminary report after 5 days of incubation, and a second final report at the end of the incubation period. 

The absence of growth from all tissues and/or prosthetic/joint fluid samples should be referred to as: “Absence of growth”. 

## 4. Conclusions

Given the complexity of prosthetic device infections, there are several critical aspects that influence laboratory diagnosis. In particular, processing methods of the biological samples, as well as the patient’s clinical information, if not addressed and carefully considered, may lead to an incorrect or missed diagnosis, influencing patient and healthcare outcomes. Figure 1 summarizes the 10 recommendations/critical issues which should be followed for a correct diagnosis of PJIs and implant related infections.

The isolation of the same microorganism from at least two samples is a major criterion for PJI. The isolation of one microorganism in a single sample may still represent infection. The distinction between pathogen and contamination is one of the most challenging steps in the diagnostic algorithm. Consequently, the method of sample collection by the orthopaedic specialist is of paramount importance to avoid the risk contamination. Contamination of prosthetic and tissues samples potentially accounts for 3% to 52% of false positive results [78]. The main sources of contaminating microorganisms include the environment of the operating room, despite the presence of an appropriate ventilation control, and through contact of the samples with other surfaces before the processing. A recent study has shown direct (operating bed) insertion of biological materials in a suitable sterile containers significantly reduces the percentage of contaminating microorganisms, compared to transfer of the samples on a tray outside the surgical field, or transferring them from the first operator to other personnel before placing the samples in sterile containers [84]. Collection of tissue specimens have higher rates of pathogen detection and reduced risk of contamination compared with swabs (classic or flocked) [41,42]. The diagnostic yield of joint aspirate directly relates to the quantity collected; this is particularly the case for aspiration of hip joints, even with the aid of ultrasound guidance. In such cases, the instillation and withdrawal of saline may improve the yield however leukocyte count cannot be performed on specimens obtained through this approach [85].

The number of specimens obtained is critical for accurate diagnosis of peri-prosthetic and implant-related infections. Between three and five tissues samples should be collected [4]. In a recent study the collection of five tissue specimens was associated with the highest accuracy for diagnosis of peri-prosthetic joint infection [86]. When homogenization and inoculation of tissue specimens in blood culture bottles was performed, fewer samples (three tissues specimens) was adequate [51]. Other studies have suggested five samples should be collected, however this was dependent on the type microorganisms and the site of infection [78,79,80,81,82,83,86]. Conversely, a larger number of samples may also increase the risk of contamination [84]. The optimal number of specimens remains an issue of controversy in the literature. Further studies including novel approaches such as molecular methods, sterile collection procedures or new ways of sampling, are required to definitively address this knowledge gap. 

In conclusion, the diagnosis of prosthetic joint infection involves the cooperation between several specialists. Current evidence has demonstrated improved patient outcomes when PJIs are managed by a multidisciplinary team with established diagnosis and management algorithms [87]. Diagnostic approaches aim to establish whether the joint is infected firstly, and then to define the involved microorganism. These approaches should incorporate accurate clinical evaluation of the patient, specific pre-operative investigations, such as inflammatory markers and, where indicated, synovial fluid aspiration for microbiological diagnosis. The collection of multiple peri-prosthetic tissues is important. In addition, processing of the implant using methods to dislodge the biofilm may also assist with microorganism detection. Cultures should be always performed in an accredited clinical microbiology laboratory and should include cultures for both aerobic and anaerobic organisms. 

## Figures and Tables

**Figure 1 jcm-08-00933-f001:**
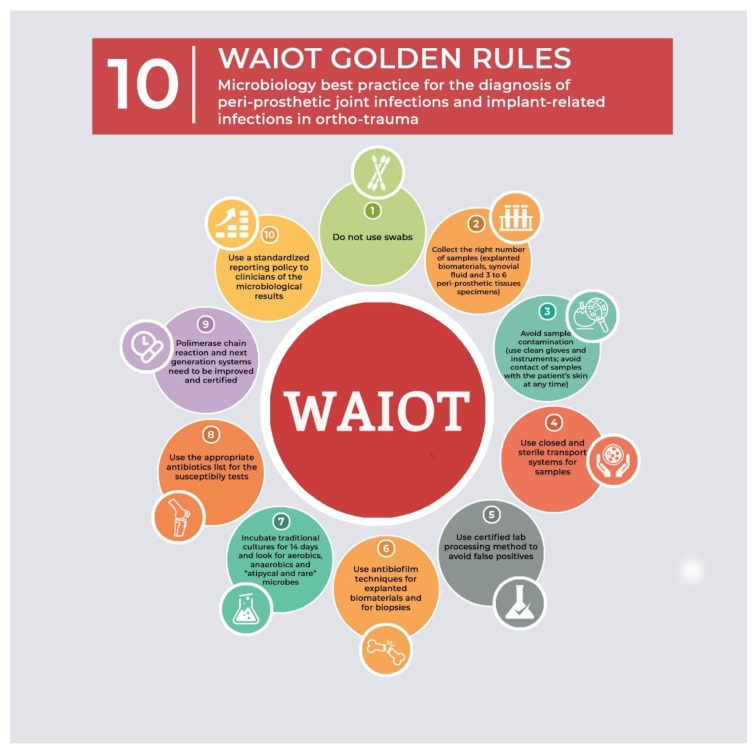
Microbiology best practice for the diagnosis of peri-prosthetic joint infections and implant-related infections in ortho-trauma. The 10 WAIOT golden rules.

**Table 1 jcm-08-00933-t001:** The key role played by the microbiological diagnosis (highlighted in bold) in the most commonly adopted peri-prosthetic joint infection (PJI) definitions, published from 2011 to 2018 [9]. Muscoloskeletal Infection Society (MSIS); Infectious Diseases Society of America (IDSA); International Consensus Meeting (ICM); European Bone and Joint Society (EBJIS).

Definition Source	MSIS 2011 [1]	IDSA 2013 [2]	ICM 2013 [3]	ICM 2018 [4]	Proposed EBJIS 2018 [5]
**Scoring system**	1 of the 2 Major Criteria OR ≥4 of 6 Minor Criteria *	≥1 Positive Criteria *	1 of the 2 Major Criteria OR ≥3 of 5 Minor Criteria *	1 of the 2 Major Criteria OR Minor criteria scoring ≥6 Infected 3–5 Possibly infected (“Consider further molecular diagnostics such as next-generation sequencing”) <3 Not infected *	≥1 Positive Criteria
	* “PJI may be present if fewer than four of these criteria are met”	* “The presence of PJI is possible even if the above criteria are not met (…)”	* “PJI may be present without meeting these criteria, (…).”	* “Proceed with caution in adverse local tissue reaction, crystal deposition disease, slow growing organisms”	
**Criteria**	Major: 1. Sinus tract communicating with the prosthesis; **2.** **A pathogen is isolated by culture from at least two separate tissue or fluid samples obtained from the affected prosthetic joint** Minor: (a) Elevated ESR (>30 mm/h) and CRP (>10 mg/L) concentration (b) Elevated synovial leukocyte count (c) Elevated PMN% (d) Purulence in the affected joint **(e)** **Isolation of a microorganism in one culture of periprosthetic tissue or fluid** (f) Greater than five neutrophils per high-power field in five high-power fields observed from histologic analysis of periprosthetic tissue at ×400 magnification	1 Sinus tract communicating with the prosthesis 2 Purulence without other etiology surrounding the prosthesis 3 Acute inflammation seen on histopathological examination of the periprosthetic tissue **4.** **≥2 intraoperative cultures or combination of preoperative aspiration and intraoperative cultures yielding an indis tinguishable organism (the growth of a virulent microorganism (e.g., *Staphylococcus aureus*) in a single specimen of a tissue biopsy or synovial fluid is also considered as indicative of a PJI)**	Major 1. A sinus tract communicating with the joint **2.** **Two positive periprosthetic cultures with phenotypically identical organisms,** Minor: (a) Elevated ESR (>30 mm/h) and CRP (>100 mg/L for acute infections; >10 mg/L for chronic infections) (b) Elevated synovial fluid WBC count (>10,000 cells/mL for acute infections; >3000 cells/mL for chronic infections) or ++ change on leukocyte esterase test strip (c) Elevated PMN% (>90% for acute infections; >80% for chronic infections) (d) Positive histological analysis of periprosthetic tissue (>5 neutrophils per high-power field in five high-power fields observed on periprosthetic tissue at ×400 magnification) **(e)** **A single****positive culture**	Major: 1. Sinus tract with evidence of communication to the joint or visualisation of the prosthesis **2.** **Two positive growths of the same organism using standard culture methods** Minor: (a) Elevated CRP (>100 mg/L for acute infections; >10 mg/L for chronic infections) or D-Dimer (unknown threshold for acute infection; >860 µg/L for chronic infection) (score 2) (b) Elevated ESR (no role for acute infections; >30 mm/h for chronic infections) (score 1) (c) Elevated synovial WBC count (>10,000 cells/mL for acute infections; >3000 cells/mL for chronic infections) OR Leukocyte Esterase (++ for acute and chronic infections) OR Positive alpha-defensin (score 3) (d) Elevated synovial PMN% (>90% for acute infections; >70% for chronic infections) (score 2) **(e)** **Single positive culture (score 2)** (f) Positive histology (score 3) (g) Positive intraoperative purulence (score 3)	1. Purulence around the prosthesis or sinus tract 2. Increase synovial fluid leukocyte count (>2000 cells/mL or >70% granulocytes) 3. Positive histopathology **4.** **Confirmatory microbial growth in synovial fluid, periprosthetic tissue, or sonication culture** **(“Confirmatory microbial growth in periprosthetic tissue: if positive in ≥1 specimen in highly virulent organisms or ≥2 in low virulent pathogens; sonication culture considered positive if >50 colony-forming units/mL of sonication fluid.”)**

**Table 2 jcm-08-00933-t002:** The key role played by the microbiological diagnosis (highlighted in bold) in WAIOT definition of peri-prosthetic joint infection (PJI), proposed in 2019 [5]. Biofilm-related implant malfunction (BIM). Low-Grade PJI (LG-PJI). High-Grade PJI (HG-PJI).

	No Infection	Contamination	BIM	LG-PJI	HG-PJI
Clinical presentation	One or more condition(s), other than infection, can cause the symptoms or the reason for reoperation (e.g., wear debris, metallosis, recurrent dislocation or joint instability, fracture, malposition, neuropathic pain)		One or more of the following: otherwise “unexplained” pain, swelling, stiffness		Two or more of the following: pain, swelling, redness, warmth, functio laesa
# of Positive Rule IN minus # of Negative Rule OUT tests	<0	<0	<0	≥0	≥1
**Post-operatively confirmed if**	**Negative cultural examination**	**One pre- or intra-operative positive culture**, with negative histology	**Positive cultural examination (preferably with antibiofilm techniques)** and/or positive histology		

Abbreviations: WAIOT: World Association against Infection in Orthopedics and Trauma; BIM: Biofilm-related Implant malfunction; LG-PJI: Low-Grade Peri-Prosthetic Joint Infection; HG-PJI: High-Grade Peri-Prosthetic Joint Infection.

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
