# Peer review of "The World Association against Infection in Orthopaedics and Trauma (WAIOT) procedures for Microbiological Sampling and Processing for Periprosthetic Joint Infections (PJIs) and other Implant-Related Infections†"

_jcm, 2019, doi:10.3390/jcm8070933_

Round 1
Reviewer 1 Report
It is a well described paper about how to manage diagnosis of osteoarticular infections even if there is nothing new. It is always good the repeat the good procedure when we see the long way for many patients to the good diagnosis and the optimal treatment.
Author Response
We are glad that this Reviewer have had good feeling with the manuscript. Thus, we would like to thank him/her for the kind words used for the judgment of the manuscript.
Reviewer 2 Report
The manuscript is a Guideline for microbiological diagnosis from the WAIOT, based in the protocols of the Italian Association. The article is interesting, and is a good review in this field. Some comments must be made trying to improve the manuscript's quality:
1-Introduction: Recent reviews must be considered, such as the 2nd International Consensus, published this year in the Journal of Arthroplasty, instead of older reviews.
2-Lines 82-86: This paragraph is true for chronic infections. However, acute ones are more "planktonic", and the importance of biofilms is minimal. The implications of this fact in diagnosis must be included in the text (interpretation of cultures, number of positive cultures, etc.)
3-Lines 244-249: A reverence fot the different number of incubation days between aerobic and anaerobic cultures (solid media) is required here. Moreover, there are studies that reported 7 days (not 14) as a good time to obtain almost all diagnosis, and probably need some comments here.
4-Lines 273-280: It is true that the sonication protocol by Trampuz et al. does not include centrifugation, but others did it (See Tande et al Clin Microbiol Rev 2014; 27: 302). This must be reviewed in the text.
5-Molecular methods: Lines 321-324: It is true that 16s rDNA homemade PCRs could have an unacceptable false positve rate. However, this assertion cannot be applied to other molecular biology approaches, especially those based in multiplex-PCR or microarrays, which have specificities higher than 95 % (even higher than 98 % among patients with a high spuspicion index). So, this phrase is false as it is written, and must be changed in any case. There are enough references that support the fact that some PCR approaches are a good methodology for PJI diagnosis.
Author Response
1-Introduction: Recent reviews must be considered, such as the 2nd International Consensus, published this year in the Journal of Arthroplasty, instead of older reviews.
Answer
As requested by the Reviewer we have now added this more recent paper: Parvizi J, Tan TL, Goswami K, Higuera C, Della Valle C, Chen AF, Shohat N. The 2018 Definition of Periprosthetic Hip and Knee Infection: An Evidence-Based and Validated Criteria. J Arthroplasty. 2018 May;33(5):1309-1314.
2-Lines 82-86: This paragraph is true for chronic infections. However, acute ones are more "planktonic", and the importance of biofilms is minimal. The implications of this fact in diagnosis must be included in the text (interpretation of cultures, number of positive cultures, etc.).
Answer:
Thank you for having underlined this aspect. Now this concept has been added on the text.
3-Lines 244-249: A reverence fot the different number of incubation days between aerobic and anaerobic cultures (solid media) is required here. Moreover, there are studies that reported 7 days (not 14) as a good time to obtain almost all diagnosis, and probably need some comments here.
Answer
Thank you for this additional point. This paragraph seems clear in regards to the incubation time, and distinguishes between aerobic and anaerobic cultures, especially when blood culture automated systems are used. However, this part has been improved by adding reference reporting 7 days as optimal time of incubation.
4-Lines 273-280: It is true that the sonication protocol by Trampuz et al. does not include centrifugation, but others did it (See Tande et al Clin Microbiol Rev 2014; 27: 302). This must be reviewed in the text.
Answer
Thank you for this nice support. Centrifugation step is now added with the related reference.
5-Molecular methods: Lines 321-324: It is true that 16s rDNA homemade PCRs could have an unacceptable false positve rate. However, this assertion cannot be applied to other molecular biology approaches, especially those based in multiplex-PCR or microarrays, which have specificities higher than 95 % (even higher than 98 % among patients with a high spuspicion index). So, this phrase is false as it is written, and must be changed in any case. There are enough references that support the fact that some PCR approaches are a good methodology for PJI diagnosis.
Answer
Thank you for these additional information. The text has been revised accordingly and many references are now added in order to clarify this aspect.
Reviewer 3 Report
Thank you for giving me the opportunity to revise this nice narrative review.
Although the topic is interesting and potentially helpful for the reader, I have some major concerns.
In particular, I believe that most of the references are not recent. For example, I would include recent findings that might contribute to the diagnosis in the near future (Yermak et al, J Infect, 2019, Karczewski et al, Bone Joint J 2019).
Also, I would discuss the use of sonication in other fields such as spine surgery (Prinz et al, J Neurosurg Spine 2019), cement spacers (Sambri et al Orthopedics 2019) and megaprosthesis (Sambri et al J Microbiol Methods 2019)
I believe that too many self-citations of the Authors are included. For example, there are more recent and more powerful studies on the use of dithiothreitol (Sambri et al Clin Orthop Relar Res 2018).
The paper need to be checked by an English native speare, too many grammar errors.
Author Response
Thank you for giving me the opportunity to revise this nice narrative review.
Although the topic is interesting and potentially helpful for the reader, I have some major concerns.
In particular, I believe that most of the references are not recent. For example, I would include recent findings that might contribute to the diagnosis in the near future (Yermak et al, J Infect, 2019, Karczewski et al, Bone Joint J 2019).
Answer
More recent papers have been now added in the different sessions (see the text), including also those suggested by the reviewer. However, the paper “Performance of synovial fluid D-lactate for the diagnosis of periprosthetic joint infection: A prospective observational study.” By Yermark et al. is not included in the text because out of the scope of this review. The review is indeed focused only on the Microbiological procedures and not on other biochemical markers. The paper of Krczewski et at has been properly included in the discussion as suggested by the Reviewer.
Also, I would discuss the use of sonication in other fields such as spine surgery (Prinz et al, J Neurosurg Spine 2019), cement spacers (Sambri et al Orthopedics 2019) and megaprosthesis (Sambri et al J Microbiol Methods 2019)
Answer
We would like to thank the Reviewer for these useful inputs. We now add and comment these articles in the text.
I believe that too many self-citations of the Authors are included. For example, there are more recent and more powerful studies on the use of dithiothreitol (Sambri et al Clin Orthop Relar Res 2018).
Answer
The paper mentioned by the reviewer is really interesting and deserves to be added and discussed. Many thanks for this additional and very useful information.
The paper need to be checked by an English native speare, too many grammar errors.
Answer
The paper had been previously checked and revised by a native English speaker (dr Trisha Peel, who is one of the co-author), but as suggested by the Reviewer a more deeper look inside the manuscript has been now furtherly performed.
Round 2
Reviewer 2 Report
OK as is
Reviewer 3 Report
Thank you for the great efforts made to ameliorate the paper.
I think that the paper is almost ready to be accepted. However, the Authors replied that they added a reference on the sonication of megaprosthesis in oncologic patients, which is not in the list (Sambri et al, J Microbiol Methods 2019). Did they forget?
Should be added before acceptance.